# Mechanistic Intimate Insights into the Role of Hydrogen Sulfide in Alzheimer’s Disease: A Recent Systematic Review

**DOI:** 10.3390/ijms242015481

**Published:** 2023-10-23

**Authors:** Constantin Munteanu, Daniel Andrei Iordan, Mihail Hoteteu, Cristina Popescu, Ruxandra Postoiu, Ilie Onu, Gelu Onose

**Affiliations:** 1Faculty of Medical Bioengineering, University of Medicine and Pharmacy “Grigore T. Popa”, 700454 Iași, Romania; elipetromed@yahoo.com; 2Teaching Emergency Hospital “Bagdasar-Arseni” (TEHBA), 041915 Bucharest, Romania; hoteteu@yahoo.com (M.H.); postoiu.ruxandra@yahoo.ro (R.P.); gelu.onose@umfcd.ro (G.O.); 3Department of Individual Sports and Kinetotherapy, Faculty of Physical Education and Sport, ‘Dunarea de Jos’ University of Galati, 800008 Galati, Romania; daniel.iordan@ugal.ro; 4Faculty of Medicine, University of Medicine and Pharmacy “Carol Davila” (UMPCD), 020022 Bucharest, Romania

**Keywords:** Alzheimer’s Disease, Hydrogen Sulfide (H_2_S), amyloid-beta (Aβ) aggregation, tau hyperphosphorylation, cellular homeostasis, neuroinflammation, neuroprotection

## Abstract

In the rapidly evolving field of Alzheimer’s Disease (AD) research, the intricate role of Hydrogen Sulfide (H_2_S) has garnered critical attention for its diverse involvement in both pathological substrates and prospective therapeutic paradigms. While conventional pathophysiological models of AD have primarily emphasized the significance of amyloid-beta (Aβ) deposition and tau protein hyperphosphorylation, this targeted systematic review meticulously aggregates and rigorously appraises seminal contributions from the past year elucidating the complex mechanisms of H_2_S in AD pathogenesis. Current scholarly literature accentuates H_2_S’s dual role, delineating its regulatory functions in critical cellular processes—such as neurotransmission, inflammation, and oxidative stress homeostasis—while concurrently highlighting its disruptive impact on quintessential AD biomarkers. Moreover, this review illuminates the nuanced mechanistic intimate interactions of H_2_S in cerebrovascular and cardiovascular pathology associated with AD, thereby exploring avant-garde therapeutic modalities, including sulfurous mineral water inhalations and mud therapy. By emphasizing the potential for therapeutic modulation of H_2_S via both donors and inhibitors, this review accentuates the imperative for future research endeavors to deepen our understanding, thereby potentially advancing novel diagnostic and therapeutic strategies in AD.

## 1. Introduction

Alzheimer’s Disease (AD), the predominant version of dementia, imposes a substantial public health burden [1], afflicting an estimated 50 million individuals globally [2]. While considerable strides have been made in understanding the pathological features of AD—namely, the accumulation of amyloid-beta (Aβ) peptides [3] and the hyperphosphorylation of tau proteins [4,5]—these classical hallmarks only partially account for the disease’s multifactorial etiology [6]. This has invigorated scholarly inquiries into the roles of other critical biochemical entities [7], among which Hydrogen Sulfide (H_2_S) has emerged as a molecule of a significant role [8].

Alzheimer’s Disease Assessment Scale (ADAS-Cog) is used to quantify cognitive dysfunction [9,10]. Data show a compelling correlation between higher ADAS-Cog scores and elevated levels of H_2_S metabolites in plasma [11,12].

A triad of neuropathological features characterizes AD [13]: neurofibrillary tangles composed of insoluble tau protein [14], amyloid plaques primarily made up of amyloid β protein (Aβ) [15], and vascular amyloidosis, specifically cerebral amyloid angiopathy (CAA) [16]. The amyloid cascade hypothesis posits Aβ as the pivotal factor instigating neurodegeneration and cognitive decline [17]. Aβ is derived from the cleavage of amyloid precursor protein (APP), regulated by secretase activities, and produced both systemically and by neurons [18]. Its clearance across the blood–brain barrier and accumulation in vessel walls present a complex interplay, suggesting a failure of Aβ elimination as a crucial factor in AD pathogenesis [19]. 

Apolipoprotein E (ApoE), particularly the ε4 isoform, serves as a significant genetic risk factor, possibly by impairing Aβ clearance [20]. Familial AD tends to involve mutations in APP or presenilins, leading to Aβ overproduction, whereas sporadic AD often arises from impaired Aβ clearance, influenced by age and ApoE genotype [3]. Collectively, these elements underscore the multifaceted nature of AD, where biochemical pathways, genetic factors, and impaired clearance mechanisms converge to contribute to disease progression [21,22,23].

The synergistic relationship between hypoxia and AD [24] and the modulatory role of H_2_S [25] presents an intricate domain ripe for scientific dissection. Contemporary research on AD is beleaguered by the limited comprehension of underlying etiological mechanisms, thereby necessitating rigorous examination of pathophysiological processes and potential therapeutic vectors [20]. 

Originally identified for its cytotoxicity in the early 18th century, H_2_S has undergone a radical reevaluation in scientific discourse [26]. It has been implicated in numerous neurophysiological processes, transitioning from an entity of toxicological concern to a critical regulator in cellular biochemistry and brain function [27]. Its role as a signaling and neurotransmitter molecule has expanded our understanding of its multifaceted impact on cellular physiology, including synaptic plasticity [28] and oxidative stress regulation [29].

H_2_S exhibits a concentration-dependent bifunctionality, offering both protective and deleterious effects on mitochondrial function [23]. Low concentrations of H_2_S appear to promote mitochondrial biogenesis and autophagic clearance of damaged organelles [30,31], whereas higher concentrations disrupt mitochondrial bioenergetics by inhibiting cytochrome c oxidase [25]. Hence, an unmet need exists for developing precise, sensitive methodologies capable of quantifying in vivo H_2_S concentrations across a range from nanomolar to micromolar levels [32]. This would enable the disentanglement of its dichotomous roles in physiological and pathological conditions, including neurodegenerative diseases [33].

The biochemical synthesis of H_2_S is orchestrated through the enzymatic activities of cystathionine γ-lyase (CSE), cystathionine β-synthase (CBS), and 3-mercaptopyruvate sulfurtransferase (3-MST), which are regulated by the transsulfuration pathway [34]. Their tissue-specific expression and differential regulation under stress conditions add another layer of complexity regarding their roles in neurobiology [35]. Moreover, the spatial compartmentalization of these enzymes suggests cell type-specific roles [36] that have not yet been elucidated. The observed downregulation of S-adenosyl methionine (SAM), a potential allosteric regulator of CBS, in AD patient samples [37] offers a compelling avenue for further research, as does the documented post-translational modification of crucial enzymes via sulfhydration, a process mediated by H_2_S [38].

Additional complexity emerges from the interconnectedness of dietary methionine, homocysteine levels, and the transsulfuration pathway [37]. Elevated homocysteine levels have been reported in the context of AD, implicating dietary methionine intake and subsequent metabolic pathways as potential exacerbators or even causal agents in AD pathology [39]. In the milieu of these intricate biochemical interactions, H_2_S is an intriguing nexus requiring comprehensive multidisciplinary studies to decode its multi-dimensional roles in AD pathology and potential therapeutics [40].

Beyond H_2_S, emerging research posits Reactive Sulfur Species (RSS) as potent antioxidants, regulators of redox homeostasis, and modifiers of protein polysulfides, warranting their inclusion as a focus of ongoing research [41]. Furthermore, the RSS-induced modifications in the Heat Shock Protein 90 (HSP90) chaperone system unveil another layer of complexity, raising questions about protein quality control pathways. These observations underscore the need for an in-depth investigation into the multifaceted roles of RSS in neurodegenerative conditions [42].

The signaling cascade based on p38 mitogen-activated protein kinase (MAPK) and the nuclear factor kappa-light-chain-enhancer of activated B cells (NF-κB) pathway has also been implicated in the neuroinflammatory processes associated with AD [43]. These pathways serve as conduits for the production of inflammatory cytokines, including interleukin-β (IL-1β), tumor necrosis factor-alpha (TNF-α), and interleukin-6 (IL-6), and their activation has been observed in the presence of Aβ plaques [44]. 

The intricate nexus of H_2_S and its interplay with various physiological and pathological pathways necessitates a multipronged approach involving biochemical, transcriptomic, proteomic, and computational analyses (Figure 1). By elucidating the complex interdependencies among these signaling pathways and H_2_S dynamics, a more comprehensive understanding of AD etiology may be achieved, thereby providing a foundation for the development of targeted, more effective therapeutics [45]. Therefore, the critical objective at the intersection of these intricate biochemical and cellular landscapes is to consolidate a coherent, multi-dimensional framework that can inform and guide targeted therapeutic interventions for AD. This review calls for a heightened level of interdisciplinary collaboration, which, coupled with advanced computational tools, holds the promise of unlocking novel therapeutic avenues for one of the most debilitating diseases of our time.

## 2. Methods

Our systematic review followed PRISMA guidelines and registered on PROSPERO ID: 449843. To ensure comprehensive coverage of relevant literature, we searched multiple databases, including PubMed, Scopus/Elsevier, Web of Science, and Google Academic. The search strategy used specific keywords and medical subject headings (MeSH) related to “Hydrogen Sulfide/H_2_S” AND Alzheimer’s disease. In addition to the databases mentioned above, we also searched https://clinicaltrials.gov/ (accessed on 15 September 2023). to identify potential clinical trials related to our topic, resulting in one registered trial. After removing non-eligible and duplicate references, our review included 133 relevant studies (Appendix A).

## 3. Results

The ubiquitous escalation of Alzheimer’s Disease (AD), the preeminent subtype of neurodegenerative dementia, presents a daunting conundrum for contemporary healthcare infrastructures [46]. The disease’s etiopathology is inherently complex, incorporating multifarious factors that extend beyond the traditional tenets of the “amyloid cascade hypothesis” [47]. Although this prevailing model implicates the aggregation of amyloid-beta (Aβ) plaques and the formation of hyperphosphorylated tau (p-Tau) fibrils as seminal events in AD pathogenesis, pharmacological strategies targeting these entities have largely failed to yield transformative clinical outcomes [48]. In light of emerging data, there is a growing consensus that amyloidogenic processes are not isolated phenomena but interact synergistically with cerebrovascular dysfunctions [49]. Thus, a critical reassessment of the vascular facets involved in AD’s complex pathophysiological architecture is imperative [50].

### 3.1. Dysregulation of H_2_S Metabolism and Vascular Pathology in AD

A confluence of findings from advanced molecular biology and proteomic analyses illuminates pronounced aberrations in the enzymatic machinery responsible for H_2_S metabolism, particularly cystathionine β-synthase (CBS) and 3-mercaptopyruvate sulfurtransferase (3-MST), in the postmortem cerebral tissue of AD patients vis-a-vis age-matched controls. This epidemiological synchronicity, observed in parallel with a global surge in both neurodegenerative and cerebrovascular disorders, underscores the contributory role of vascular risk factors modifiable through socioeconomic interventions [7].

The emergent “vascular dysregulation hypothesis” postulates a consequential disequilibrium between cerebral perfusion and the metabolic exigencies of neural tissue, thereby amplifying vulnerability to a panoply of AD-associated risk factors. These encompass hypertension, underlying cerebrovascular pathologies, and lifestyle determinants such as physical inactivity [12]. Vascular aberrations endemic to AD have garnered considerable academic scrutiny, with a particular focus on how H_2_S dysregulation engenders vascular irregularities like endothelial dysfunction and diminished cerebral blood flow—variables that exhibit a consistent correlative relationship with AD severity [51,52,53].

In-depth mechanistic studies have begun to elucidate the intricate pathways through which vascular dysregulation exacerbates existing neurodegenerative processes in AD [54]. Vascular imbalances are implicated in initiating a pro-inflammatory cerebral milieu, compromising the structural and functional integrity of the blood–brain barrier (BBB) [55]. This disintegrity triggers a downstream cascade of biochemical perturbations, encompassing excitotoxic calcium signaling mechanisms (often referred to as the “calcium hypothesis”) and metabolic disruptions, culminating in the aggregation of amyloid-beta plaques and hyperphosphorylated tau (p-Tau) fibrils. Such mechanistic complexities further obfuscate the traditional diagnostic demarcations separating AD from vascular dementia, thereby accentuating their interconnected etiopathological origins [56]. This paradigm shift has not gone unnoticed; the ”National Institute on Aging and the Alzheimer’s Association” have recently endorsed the integration of vascular biomarkers into comprehensive diagnostic panels for AD [57].

Endogenous plasma H_2_S, a ubiquitous gasotransmitter instrumental in vascular homeostasis, is increasingly recognized as a viable vascular biomarker intricately associated with cognitive impairment and disease progression in AD [58,59]. 

The burgeoning research landscape now explores the homeostatic and regulatory influences of H_2_S and its metabolites on both vascular and neuronal compartments. Functioning as a pleiotropic effector molecule, H_2_S orchestrates a wide array of physiological processes pivotal for maintaining vascular integrity and neuronal vitality [60]. Therefore, this integrative narrative seeks to recalibrate our conceptualization of AD, repositioning it not as a monolithic neurodegenerative entity but as a complex interplay of vascular and neuronal dysfunctions, with plasma H_2_S emerging as both a nuanced biomarker and a prospective therapeutic agent.

### 3.2. The Dialectic Role of Hydrogen Sulfide in Neurovascular and Cognitive Dysregulation in Alzheimer’s Disease

While existing literature has robustly established the role of H_2_S as a circulatory biomarker for vascular disorders [61], burgeoning research is increasingly elucidating its complex roles within neurovascular compartments, particularly concerning Alzheimer’s Disease (AD) [62]. Emerging investigations have transcended the focus on H_2_S solely as a modulator of amyloid-beta (Aβ) aggregation dynamics in neuronal cultures, delving into its function in regulating the transvascular clearance of Aβ peptides [63]. A correlative relationship between diminished H_2_S levels and compromised blood–brain barrier integrity has been observed, thereby exacerbating Aβ accumulation [64]. In vitro assays indicate that restoration of H_2_S levels via exogenous donors ameliorates vascular deficiencies and mitigates Aβ-induced cytotoxic effects. In contrast, other studies have posited a correlation between elevated plasma H_2_S levels and AD pathogenesis [65].

Functionally, H_2_S assumes dual and somewhat paradoxical roles within biological systems [66]. It serves as a neuromodulatory signaling molecule that interacts with neurotransmitter receptors, notably the NMDA receptor and is involved in long-term potentiation processes, contributing to neuroprotection, cognitive function, and memory consolidation [67]. Conversely, elevated H_2_S concentrations and metabolites can exert neurotoxic effects, contributing to neuronal stress and vascular dysregulation [25]. This bifunctionality resonates with observations related to other gasotransmitters like nitric oxide, suggesting a homeostatic balance susceptible to pathological deviation under certain conditions [23].

Within the central nervous system, H_2_S biosynthesis occurs across various cellular compartments, adding a layer of biochemical complexity. In the brain parenchyma, cystathionine β-synthase (CBS) serves as the principal enzyme for H_2_S generation, while cystathionine γ-lyase (CSE) is responsible for H_2_S production within cerebral microvessels. Further intricacy is introduced by the existence of multiple biochemical forms of H_2_S, encompassing free H_2_S, acid-labile sulfides, and bound forms such as persulfides and polysulfides [35].

An analytical review of the extant data implicates H_2_S as a significant factor in the correlation between cognitive dysfunction and white matter lesions [12], a salient marker of microvascular pathology [68]. Classifier analyses underscore the diagnostic utility of total plasma sulfide burden as an accurate discriminator between AD and control states [69]. Moreover, one mechanistic pathway through which elevated H_2_S levels may be implicated in AD emerges from the “vascular dysregulation hypothesis”, which posits that cerebral hypoxia contributes to AD etiology. Hypoxic and ischemic conditions are known to induce CSE expression, potentially resulting in increased H_2_S biosynthesis under compromised oxygenation, especially in aging brains [12].

The amalgamation of these findings presents compelling evidence for the multifaceted role of H_2_S as both a diagnostic biomarker and prospective therapeutic target in the complex landscape of AD. The data highlight intricate interactions between H_2_S and diverse neurovascular and cognitive pathways that warrant further investigation. Future inquiries should aim to dissect the modulatory influences of H_2_S across various pathological states, explore the pharmacodynamics of H_2_S-targeted interventions, and assess the clinical implications of these insights within the broader rubric of multimorbid neurovascular diseases [8].

### 3.3. In-Depth Analysis of H_2_S’s Role in Inflammation

Neuroinflammation, sustained by activation of microglia and astrocytes with concomitant release of pro-inflammatory cytokines, is a hallmark of AD [70]. Altered H_2_S signaling can significantly impact the inflammatory landscape in the AD brain [43,71]. 

The multifaceted mechanisms through which H_2_S orchestrates inflammatory pathways include:H_2_S impedes the activation of the nuclear factor kappa-light-chain-enhancer of activated B cells (NF-κB), a central transcription factor in inflammatory responses [72,73]. By preventing the translocation of NF-κB to the nucleus, H_2_S curtails the expression of downstream pro-inflammatory cytokines like TNF-α, IL-1β, and IL-6, which exacerbate neuroinflammation and contribute to neuronal damage in AD [74,75].H_2_S promotes mitochondrial biogenesis and fuses while inhibiting mitochondrial fission. This optimization of mitochondrial dynamics diminishes the release of mitochondrial DNA and other DAMPs (damage-associated molecular patterns) that can elicit inflammatory responses [60].H_2_S can influence the activity of various ion channels, including calcium [76], potassium [77], and TRP channels [78], thus modulating intracellular signaling pathways linked to inflammation [79].Given that ROS can activate various inflammatory cascades, the ROS-scavenging property of H_2_S offers an indirect anti-inflammatory effect. Diminished H_2_S levels can accentuate oxidative stress, which can further stoke the flames of neuroinflammation, potentiating the neurodegenerative cascade in AD [80].

Harnessing the anti-inflammatory potential of H_2_S offers a promising therapeutic avenue for AD. Compounds like GYY4137 and sodium hydrosulfide (NaHS) can release H_2_S in a sustained manner, thereby offering anti-inflammatory benefits. Their administration can restore optimal H_2_S levels, attenuating neuroinflammation [81].

By targeting enzymes like cystathionine β-synthase (CBS) and cystathionine γ-lyase (CSE), which are involved in H_2_S biosynthesis, it is possible to regulate endogenous H_2_S levels, thus modulating the inflammatory response. Hybrid molecules that can release H_2_S while also exerting other anti-inflammatory effects (e.g., H_2_S-releasing NSAIDs) can offer synergistic benefits in dampening neuroinflammation [82].

### 3.4. Connection between H_2_S and Oxidative Stress in AD

Oxidative stress is a cardinal feature of AD, and perturbations in H_2_S signaling can substantially influence the oxidative milieu in the AD brain [83]. H_2_S, emerging as a pivotal regulator of cellular redox homeostasis, deploys an array of mechanisms to counteract oxidative stress at the molecular level:H_2_S can neutralize various ROS, including hydroxyl radicals and hydrogen peroxide, effectively circumventing oxidative damage to cellular components. Diminished H_2_S levels in AD impede its direct ROS-neutralizing activity, culminating in elevated ROS levels and consequent oxidative damage to lipids, proteins, and nucleic acids [84].Altered H_2_S signaling can attenuate the expression and activity of antioxidant enzymes, leaving neurons vulnerable to ROS-mediated insults. H_2_S enhances the expression and activity of endogenous antioxidant enzymes, including superoxide dismutase (SOD), glutathione peroxidase (GPx), and catalase. By bolstering these enzymatic defense systems, H_2_S fortifies cellular resistance against oxidative insults [85].H_2_S promotes the synthesis of glutathione (GSH), a critical intracellular antioxidant, ensuring that cells have ample GSH to quench ROS and repair oxidatively damaged biomolecules [86,87].By optimizing mitochondrial electron transport chain (ETC) efficiency and reducing electron leakage, H_2_S minimizes the formation of mitochondrial ROS, curtailing a significant source of intracellular oxidative stress [88].

Slow-releasing H_2_S donors, like GYY4137 [89,90], which are agents that can upregulate the activity or expression of H_2_S-producing enzymes, can elevate brain H_2_S levels, bolstering the direct and indirect antioxidative mechanisms to counteract oxidative stress in AD [91]. Also, the development of hybrid molecules that can release H_2_S and also possess intrinsic antioxidant activity can offer a dual antioxidative thrust, potentially yielding synergistic benefits in quelling oxidative stress in AD [92,93].

### 3.5. Blood–Brain Barrier Disturbances

The brain’s microvasculature serves as a robust gateway, the blood–brain barrier (BBB), that protects the neural milieu from potential toxins [94]. Abnormal homeostasis of H_2_S can compromise the BBB’s functional integrity [25,95]. A leaky BBB allows peripheral immune cells and pro-inflammatory mediators to infiltrate the brain, amplifying the neuroinflammatory response. Previous studies have indicated that polysulfide donors, not free sulfides, are implicated in weakening the vascular endothelial barriers in vitro [96]. Specifically, these compounds were shown to disrupt the BBB in the early aftermath of ischemic stroke, highlighting the damaging role of H_2_S metabolites in these critical structures. CSE, an enzyme primarily localized within the vascular network, was investigated for its contributory role in maintaining basal vascular integrity [97]. 

CSE-deficient murine models displayed enhanced barrier function both in the brain and lungs, evidenced by decreased permeability to small solutes like sodium fluorescein. These findings suggest that CSE-derived H_2_S may be a key regulator of vascular integrity, and its imbalance could very well be a nexus in AD pathology [98]. 

In the context of AD, BBB disturbances are not merely vascular issues but extend to neurobiological consequences. Disruptions in BBB may facilitate the influx of excitotoxic substances into the brain interstitium, including neurotransmitters, immunomodulatory components, and free iron ions. These can culminate in calcium-dependent excitotoxicity, corroborating the “calcium hypothesis,” thereby providing a mechanistic pathway linking vascular dysfunction to cognitive impairments observed in AD [99]. 

### 3.6. Sulfide Stress in AD

The term “sulfide stress” is introduced to articulate the complex array of metabolic and biochemical perturbations associated with elevated levels of H_2_S and its metabolites. This construct emerges as a pivotal determinant in the pathology of AD, offering nuanced insights with implications for diagnosis, prognosis, and mechanistic understanding. Early identification of sulfide stress could yield invaluable temporal frameworks for targeted interventions aimed at fortifying the blood–brain barrier (BBB), thereby retarding AD progression [100,101].

### 3.7. H_2_S, Tau Pathology, and Vascular Integrity

Sophisticated electrophoretic and immunohistochemical assays substantiate the negative correlation between H_2_S levels and the hyperphosphorylation of tau proteins, a key pathological feature of AD [102]. These investigations concurrently delve into vascular motility, revealing that diminished H_2_S levels are associated with impaired vasodilation and compromised vascular integrity [103,104]. Such findings posit H_2_S as an instrumental regulator in neuronal and vascular pathological processes within the AD milieu [105,106].

Tau proteins, primarily involved in stabilizing microtubules in neurons, undergo pathological hyperphosphorylation in AD [4,5]. This process results in their disassociation from microtubules, destabilizing the neuronal cytoskeleton and ultimately contributing to the formation of neurofibrillary tangles [107]. Reduced levels of H_2_S have been correlated with increased enzymatic activity of kinases like GSK-3β, which directly phosphorylate tau proteins [25,108]. This suggests a potential mechanistic insight: H_2_S may serve as an inhibitory modulator for kinases involved in tau phosphorylation, thus functioning as a neuroprotective agent [35].

The role of H_2_S extends beyond the neuronal milieu to encompass vascular integrity [109]. Investigations utilizing techniques like wire myography reveal that H_2_S acts as a potent vasodilator by modulating the activity of ion channels, specifically the ATP-sensitive potassium channels in vascular smooth muscle cells [110,111]. A decrease in H_2_S levels leads to impaired vasodilation, possibly by reducing the bioavailability of nitric oxide (NO), a key molecule in vascular relaxation [112]. Furthermore, diminished H_2_S levels are associated with the deterioration of endothelial function, contributing to a compromised vascular architecture [113].

This collection of findings articulates a bidirectional relationship between the H_2_S signaling pathway and pathophysiological mechanisms operative in AD. On one hand, H_2_S appears to act as a regulatory checkpoint for tau phosphorylation, which could potentially slow or prevent the formation of neurofibrillary tangles [114]. On the other hand, its role in maintaining vascular integrity may have significant implications for cerebral blood flow and, by extension, for the metabolic sustenance of neuronal tissues [25,35,115].

The orchestrated dysregulation of both neuronal and vascular systems mediated by H_2_S underscores the molecule’s multifaceted role in AD pathology. Impaired vasodilation could exacerbate neurodegenerative processes by compromising the access to oxygen and other vital nutrients of the brain, further influencing the neuronal milieu where tau pathology manifests [116]. Therefore, understanding H_2_S as an instrumental regulator provides a nuanced mechanistic framework for targeted therapeutic interventions aimed at both the neuronal and vascular aspects of Alzheimer’s Disease [117].

### 3.8. Persulfidation and Glycogen Synthase Kinase-3β in AD

Glycogen synthase kinase-3β (GSK-3β), a serine/threonine kinase, has been connected to AD pathogenesis due to its capability to phosphorylate tau proteins, facilitating the formation of neurofibrillary tangles [118]. Intriguingly, H_2_S is postulated to modulate GSK-3β activity via persulfidation, a highly orchestrated post-translational modification. A deficiency in persulfidation research within the AD context necessitates a focused inquiry into GSK-3β’s persulfidation status, potentially illuminating alternative regulatory mechanisms [119,120,121].

### 3.9. Analytical Approaches to Persulfidation

Given the scarcity of persulfidation-focused research, examining the direct persulfidation status of GSK-3β could offer seminal insights into regulatory mechanisms influencing tau pathology [122]. Notably, diminished levels of persulfidation have been documented in AD animal models and postmortem human brain tissues [29]. These attenuated persulfidation levels potentially contribute to GSK-3β hyperactivity, exacerbating tau phosphorylation [123,124]. Employing mass spectrometry to map the specific cysteine residues subject to persulfidation could catalyze groundbreaking revelations. Furthermore, targeted proteomics could quantify persulfidation degrees under diverse physiological conditions, thereby enriching our understanding of this intricate regulatory network [125].

### 3.10. Nutritional Modulation via Methionine Restriction and H_2_S Pathways

The complex relationship between dietary methionine restriction (MR), H_2_S pathways, and AD necessitates an interdisciplinary investigation encompassing cellular, systemic, and clinical domains [126]. Noteworthy is the observation that reduced methionine intake correlates with enhanced cognitive function, particularly in subjects manifesting mild cognitive impairment (MCI). However, discernable sexual dimorphisms within MR-treated AD mouse models necessitate caution when extrapolating these results to human pathophysiology [127,128].

The H_2_S pathway, orchestrated by enzymes such as cystathionine β-synthase (CBS) and cystathionine γ-lyase (CGL), plays a pivotal role in neuroprotection [7]. MR has been shown to robustly activate the CBS/H_2_S pathway in male AD model mice, mitigating oxidative stress and enhancing mitochondrial biogenesis [129]. This sexual dimorphism raises compelling questions about the translational fidelity of animal models to human disease manifestations and suggests the exigency for multifactorial, sex-specific therapeutic strategies [127,128,130].

### 3.11. Sulfur-Mediated Pathways: The Gut–Microbiota–Brain Axis

One of the points emanating from this comprehensive review is the putative synergistic relationship between sulfur metabolism—particularly that of H_2_S—and the microbiota–gut–brain axis [71]. This axis is ostensibly implicated in modulating various central nervous system (CNS) functions, including, but not limited to, the incitement of neuroinflammatory processes and the deposition of amyloid-beta aggregates in Alzheimer’s Disease (AD). Multiple therapeutic modalities, ranging from dietary interventions to faecal microbiota transplantation, have exhibited potential in reconfiguring gut microbiota, thereby mitigating the pathogenesis of neurodegenerative maladies. Within this context, H_2_S emerges as a novel interacting entity that may significantly influence the etiopathogenesis of AD, thereby expanding the horizons for future therapeutic strategies [131,132].

This exhaustive review accentuates the intricate nexus between gut microbiota and the central nervous system, particularly emphasizing the gut–brain vascular axis as an integral conduit in the pathogenesis of neurodegenerative and psychiatric disorders. The existing scientific data delineates how microbial imbalances and intestinal permeability, colloquially termed “leaky gut syndrome”, act as potent inducers of systemic inflammation and compromise the integrity of vascular barriers, notably the intestinal epithelial and blood–brain barriers. Such disturbances consequently potentiate neuroinflammatory cascades, impacting the CNS and distal organ systems. This gut–brain crosstalk is particularly germane to a spectrum of disorders, encompassing Alzheimer’s Disease, Parkinson’s Disease, and Depressive Disorder, and engages mechanisms such as neurotransmitter dysregulation and aberrant immune signaling [133]. 

The intricate nature of this axis provides fertile ground for the development of innovative biomarkers and therapeutics targeting microbial constituents. Emerging research underscores the pivotal role of the intestinal microbiota, often termed the “second genome” [134], in the pathogenesis and progression of AD. Utilizing next-generation sequencing technologies like 16S rRNA sequencing [135], studies have identified a bidirectional regulatory mechanism between the gut and brain, forming a microbiota–gut–brain axis. Dysbiosis in the intestinal microbiota is a risk factor for AD and exacerbates neuroinflammatory responses post-stroke, affecting long-term outcomes [136]. Quantifiable measures such as the Stroke Dysbiosis Index (SDI) [137] and levels of short-chain fatty acids (SCFAs) [138] have been proposed as potential prognostic indicators. These findings illuminate the therapeutic potential of targeting the intestinal microbiota for mitigating AD and its complications, opening avenues for innovative treatment approaches [139,140].

### 3.12. Therapeutic Strategies Based on H_2_S for Alzheimer’s Disease

Recent breakthroughs have highlighted the significance of H_2_S in neurophysiology, paving the way for innovative Alzheimer’s Disease (AD) therapies. Efforts have focused on creating molecules that produce this gasotransmitter in vivo. For instance, compounds like GYY4137 gradually release H_2_S, enhancing its neuroprotective and anti-inflammatory effects [89,141]. Furthermore, hybrid drugs combine NSAIDs’ anti-inflammatory attributes with H_2_S release, targeting dual therapeutic outcomes for AD [92]. Balneotherapeutic methods using sulfurous mineral waters and mud therapy also increase plasma H_2_S levels [131,142]. Additionally, strategies to boost endogenous H_2_S synthesis, such as enhancing H_2_S-producing enzymes like CBS and CSE, are under exploration, aiming for an optimal neurochemical environment in AD management [143].

Within neuroscience, H_2_S has gained prominence for its multifaceted neuroprotective roles. It modulates the release of neurotransmitters like glutamate, influencing synaptic plasticity and neural interactions [77]. Furthermore, H_2_S showcases anti-apoptotic capabilities by fine-tuning mitochondrial functions, impeding cytochrome c release, and curbing caspase activation, thereby preventing neuronal cell death. It also stimulates neurogenesis by promoting neural stem cell differentiation and growth, aiding brain repair [144]. H_2_S can neutralize reactive oxygen species and protect against oxidative damage, a pivotal factor in AD development. Preliminary research also indicates that H_2_S donors can mitigate cognitive impairments in AD animal models by preserving synaptic health [145].

Though promising, the therapeutic utilization of H_2_S is accompanied by certain hurdles. Correct dosage is crucial; while low to moderate amounts of H_2_S can offer therapeutic benefits, excessive levels might be harmful. Its transient nature prompts the need for either stable donors or formulations that ensure sustained release. Additionally, systemic administration of H_2_S raises concerns about potential off-target peripheral effects, underscoring the importance of devising targeted delivery methods [146].

As research delves deeper into the potential of H_2_S and its donors for AD treatment, various clinical trials are underway. Initial phase I trials, particularly with H_2_S-releasing compounds like GYY4137, showcased a promising safety profile, setting the stage for subsequent efficacy studies in AD patients [93,147,148]. Phase II trials are now actively evaluating the capacity of H_2_S donors to mitigate cognitive deterioration, using neuropsychological assessments as their primary indicators [25,149,150]. Moreover, given the prolonged nature of AD treatments, studies are increasingly emphasized to explore these donors’ long-term safety. While early findings indicate limited side effects, the scientific community eagerly anticipates detailed outcomes from more extensive patient groups.

## 4. Discussion

The investigation into the role of H_2_S in AD constitutes an evolving frontier that transcends the conventional confines of neurodegenerative research. As the data synthesized in the Results section demonstrate, H_2_S is not merely a peripheral player but, rather, occupies a nexus of biochemical, vascular, and cellular interactions integral to both the pathogenesis and potential therapeutics of AD [8,25].

The classical theories of AD, which predominately focus on amyloid-beta (Aβ) aggregation and tau hyperphosphorylation, have been insufficient in elucidating AD’s multifactorial etiology. The emergent role of H_2_S in modulating these hallmarks of AD serves to deepen our understanding, particularly when viewed through the lens of vascular pathology, thereby allowing for a more nuanced and holistic grasp of disease pathophysiology [35].

The discussion has to be extended into the realm of alternative therapies, such as inhalations with sulfurous mineral waters and mud therapy, which include Hydrogen Sulfide. Although these non-conventional approaches warrant further rigorous clinical validation, the preliminary evidence points towards their potential efficacy. These findings could herald a paradigm shift, introducing more holistic, multi-modal therapeutic regimens that combine conventional pharmacology with alternative therapies [131,142].

H_2_S-based interventions, employing both donors and inhibitors, have exhibited substantial promise, as substantiated by the improvement in both neural and vascular health metrics. However, the duality of H_2_S as both a pathological mediator and a protective agent complicates its therapeutic manipulation [151]. 

The intersection of nanotechnology, material science, and molecular biology has enabled the development of intelligent polymeric H_2_S delivery systems [152] with substantial clinical potential. Such systems offer precise control over H_2_S delivery and pave the way for real-time monitoring and theranostic applications. The employment of biocompatible polymers in establishing intelligent H_2_S delivery systems has ushered in a new era of sophisticated therapeutic strategies. Material selection is of paramount importance here. Molecular simulations can be employed to design optimal polymer architectures and encapsulation techniques for maximum therapeutic efficacy. Furthermore, these intelligent systems could integrate targeting ligands such as antibodies or aptamers that recognize specific molecular markers on pathological cells. This ‘lock-and-key’ mechanism enhances the targeted delivery of H_2_S, thereby reducing systemic toxicity [35].

The categorization of H_2_S donors based on their activation mechanisms is a judicious way to dissect the kinetics and dynamics of H_2_S release [153]. To augment the scope of hydrolysis-activated donors, one could explore quantum chemical methods to examine the feasibility of newly synthesized phosphorodithioate derivatives [154]. Similarly, thiol-activated donors could be studied through computational docking methods to understand their interaction with biomolecules like glutathione (GSH) [155]. Systems biology approaches could offer valuable insights into the enzyme kinetics of enzyme-activated donors, allowing for the design of donors with tailor-made enzymatic triggers [156].

While significant progress has been made, several challenges still loom large. The toxicity of donor by-products, their stability, and the potential immunogenicity of polymeric carriers remain pertinent concerns [157]. Applying proteomics and genomics could be instrumental in understanding molecular bio-interactions, thereby guiding the design of safer, more effective H_2_S delivery systems [158].

An Electrochemiluminescence (ECL) system for the selective and highly sensitive detection of H_2_S could be of help. Elevated H_2_S levels have been linked to various diseases, including Alzheimer’s and type 1 diabetes, underscoring the necessity for efficient diagnostic tools. The ECL system employs a novel redox-stable chemosensor, an iridium (III) complex (Probe) that features a “turn-on” mechanism triggered by the specific cleavage of 2,4-dinitrophenyl ether (DNP) in the presence of H_2_S. This approach fills a gap in current detection methodologies, which often suffer from high operational complexity, cost, and time consumption [159]. 

### Limitations and Therapeutic Challenges

In light of evolving pharmacological paradigms, this review serves to delineate the Mechanism of Action (MoA) as an essential parameter in the transitional trajectory from experimental research to clinical applicability. We endorse the adoption of multi-target drugs, combination therapies, and the integration of machine-learning techniques in high-dimensional ‘omics’ data analysis as quintessential strategies for potentiating the therapeutic efficacy of H_2_S in the treatment of Alzheimer’s Disease (AD) [143,160,161].

While the methodological framework of a systematic review offers a structural basis for ensuring the review’s veracity and rigor, it is also the locus where potential biases and inaccuracies can infiltrate, thereby destabilizing the integrity of the synthesis. A pre-defined, comprehensive search strategy was executed to minimize such risks; however, these measures cannot completely negate the element of bias or oversight, as they are present in scientific literature. Hence, the necessity of scrutinizing the limitations inherent to our methodological approach. The precision of the search terms and their corresponding sensitivity and specificity helped to capture the relevant literature.

The studies incorporated in this review are not of uniform significance or quality, making the equitability of their contribution to the synthesis less consistent. This undermines the homogeneity of data quality, thereby introducing an element of unknown in the ensuing analyses. Owing to the scarcity of comprehensive clinical data, a narrative synthesis was opted for instead of a more statistically robust meta-analysis. The insufficient volume of clinical data, especially on the pharmacokinetics and pharmacodynamics of H_2_S-targeted therapeutics, imposes a limitation on the extensibility of our findings to clinical settings.

## 5. Conclusions

The overarching narrative from this comprehensive systematic review underscores the complex, multifaceted role H_2_S plays in the pathophysiology and prospective treatment paradigms for Alzheimer’s Disease (AD). The traditional pathogenic models of AD, largely anchored on the role of amyloid-beta (Aβ) and tau, are incomplete and require nuanced augmentation. H_2_S emerges as a biochemically potent molecule capable of influencing Aβ and tau, as well as vascular pathology—a fact that has hitherto been underrepresented in the conventional academic discourse on AD [162].

While the neuroprotective attributes of H_2_S are incontrovertible, the molecule also exhibits a dualistic nature, with cytotoxic consequences at elevated concentrations. This paradox necessitates caution in therapeutic application and calls for more granular mechanistic studies to delineate the safe and effective dosage ranges, particularly given that current efforts remain largely at the preclinical stage [7,163].

The dualistic nature of H_2_S—acting as both a potential pathogenic factor and a possible therapeutic agent [164]—poses intricate challenges for its pharmacological manipulation. Consequently, this opens up a diverse and expansive array of research avenues, particularly in methodological domains involving systems biology, omics technologies, and computational modelling. Furthermore, the need for longitudinal studies to establish causality and temporal dynamics is an unmet research requirement. 

H_2_S dysregulation disrupts brain microvasculature, principally through its metabolites and enzymatic pathways, contributing to a vascular-neural cascade of events that manifest as cognitive impairments in AD. While preliminary, these findings warrant extensive follow-up research studies for validation. A comprehensive understanding of these mechanisms could unlock targeted pharmacological interventions that aim to restore vascular and neural homeostasis, offering promising therapeutic avenues in the battle against AD.

## Figures and Tables

**Figure 1 ijms-24-15481-f001:**
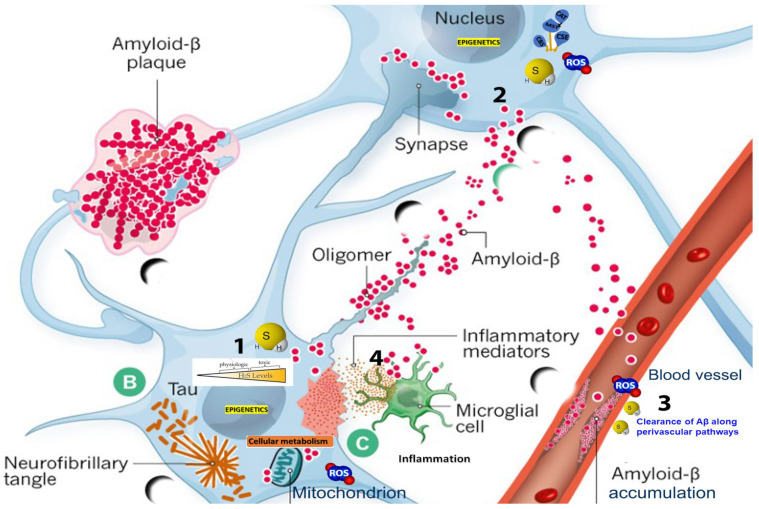
Multifaceted roles of H_2_S in Alzheimer’s Disease. 1. Biphasic, concentration-dependent effects of H_2_S, emphasizing its dual role as a toxicant at high concentrations and a cellular signaling molecule at low concentrations. 2. H_2_S’s involvement in redox homeostasis, emphasizing its protective role in mitochondrial function and regulation of oxidative stress. 3. Putative role of H_2_S in promoting the clearance of amyloid-beta (Aβ) peptides. 4. Involvement of H_2_S in modulating inflammatory pathways.

## Data Availability

Not applicable.

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
