# Peer review of "Mechanistic Intimate Insights into the Role of Hydrogen Sulfide in Alzheimer’s Disease: A Recent Systematic Review"

_ijms, 2023, doi:10.3390/ijms242015481_

Round 1

Reviewer 1 Report

The Review titled as "Mechanistic Intimate Insights into the Role of Hydrogen Sulfide in Alzheimer's Disease: A Recent Systematic Review"  focus on the function of H2S in the process of AD.

In the Abstract it said that "emphasizing the potential for therapeutic modulation of H2S via both donors and inhibitors, this review accentuates the imperative for future research endeavors to deepen our understanding thereby potentially advancing novel diagnostic and therapeutic strategies in AD.", But I don't see any summary or comments on current therapeutic research or agent by using H2S or the H2S donor.

Also The review don't talk about the  the relationship between H2S and inflammation or AD-related oxidative stress, Which is the important topics.

Need double check all the sentence and paragraph to make sure all format and spelling are correct.

Author Response

We appreciate and thank for the reviewer's observations and recognize the importance of providing a comprehensive section regarding the current therapeutic research involving H2S and its donors in the context of Alzheimer's Disease (AD).

The relationship between H2S and inflammation, as well as its role in AD-related oxidative stress, is indeed pivotal, the issues being discussed and researched more in the previous years. In the revised version of our manuscript, we have added a new subsection dedicated to elucidating the role of H2S in modulating inflammatory pathways and its implications in AD. Moreover, a subsection detailing how H2S influences oxidative stress, particularly in the context of AD, will be presented to ensure comprehensive coverage of this vital area.

In our revised manuscript, we also have included a section that provides a detailed account of the recent therapeutic advancements and ongoing research studies in this area. This section will underline the mechanisms of action, the potential benefits, and the challenges associated with H2S donors in AD therapeutic strategies.

Thank you very much for your help in improving our manuscript!

Reviewer 2 Report

The review presented by Munteanu et al and entitled “Mechanistic Intimate Insights into the Role of Hydrogen Sulfide in Alzheimer's Disease: A Recent Systematic Review” is indeed an interesting contribution to our understanding of the role of H2S in Alzheimer's disease. While this review covers several aspects comprehensively, in my opinion, it lacks a section addressing therapeutic strategies based on H2S for Alzheimer's disease.

Indeed, numerous research projects are currently underway, focused on developing molecules capable of generating this gasotransmitter in vivo. These developments represent a crucial aspect of the discussion and should be incorporated into the review.

Therefore, I kindly invite the authors to consider adding a dedicated section that delves into these therapeutic strategies and their potential implications for Alzheimer's disease.

Furthermore, I would like to suggest that Figure 2 be relocated to a supporting information section. In its current placement, it appears to have limited relevance to the main text and may be more appropriately situated in the supplementary materials of the manuscript.

Author Response

We sincerely thank the reviewer for recognizing the significance and value of our work in this area, and appreciate the feedback and acknowledge the importance of detailing therapeutic strategies rooted in H2S in the context of Alzheimer's disease. We concur that current research endeavors aimed at generating H2S in vivo are paramount in shaping the therapeutic landscape for Alzheimer's disease. We will incorporate a comprehensive section addressing the therapeutic strategies and their implications for Alzheimer's disease, as advised.

Regarding Figure 2: We will reassess its placement in relation to the main text and will relocate it to the supplementary materials as recommended.

Thank you very much for the constructive feedback and your help in improving our manuscript!

Round 2

Reviewer 1 Report

 The author added the relationship between H2S and inflammation, as well as its role in AD-related oxidative stress, which is great.

Please double check all spelling errors before submit the final version.

Reviewer 2 Report

The authors have answers to all my concerns